# The Suitability Assessment of Agricultural Drought Monitoring Indices: A Case Study in Inland River Basin

**Weiqi Liu** [1,2], **Shaoxiu Ma** [1,3,*], **Kun Feng** [1], **Yulai Gong** [1], **Linhao Liang** [1,2] **and Mitsuru Tsubo** [3]

1   Key Laboratory of Desert and Desertification, Northwest Institute of Eco-Environment and Resources, Chinese Academy of Sciences, Lanzhou 730000, China
2   University of Chinese Academy of Sciences, Beijing 100049, China
3   International Platform for Dryland Research and Education, Tottori University, Tottori 680-8550, Japan
*   Correspondence: mashaoxiu@nieer.ac.cn

**Abstract:** Drought monitoring is an important scientific basis for drought impact evaluation and the selection of mitigation strategies. Since the drivers of drought vary among regions, there is no universal drought index applicable to different regions. The Shiyang River Basin, an inland river basin, located in Gansu Province, China, has a closed water cycle system. Drought is a dominant nature disaster for the sustainable development of the region. Thus, this is an ideal area to explore the suitability of drought-monitoring indices. Here, we took the Shiyang River Basin as an example, in order to explore suitable indicators for agricultural drought monitoring in inland river basins. This study assessed the twelve different widely used drought indices used for monitoring the impact of drought on crop growth, represented by net primary production (NPP). The results showed that the vegetation status-based drought indices (VCI and NVSWI) had the highest significant correlation ($0.6 \leq |R| \leq 1$, $p < 0.05$) on NPP, and the integrated drought indices (DSI and ISDI) had the strong significant correlation ($0.4 \leq |R| < 0.6$, $p < 0.1$). These four indices are good indicators for agricultural drought monitoring. Studies based on these four indices showed that agricultural drought has a tendency to slow down from 1982 to 2020. This is inconsistent with the monitoring of drought indices based on the meteorological variables that show a trend of increasing drought. This is mainly due to the increased efficiency of water management and its use in inland river basins. This indicates that other water resource information, such as runoff, should be included to construct an integrated agricultural drought-monitoring indices in management intensive regions, such as in an inland river basin.

**Keywords:** agricultural drought; drought indices; drought assessment; inland river basin

## 1. Introduction

Drought is one of the most common natural disasters in terrestrial ecosystems, characterized by wide coverage, severe impacts and frequent occurrences, which directly or indirectly cause significant economic losses at global and regional scales [1,2]. The American Meteorological Society (AMS) classifies drought into meteorological drought, agricultural drought, hydrological drought, and socioeconomic drought [3,4]. Among them, agricultural drought refers to the impact of water shortage on the crop production, and especially when the soil water content cannot satisfy the plant requirement [5]. The latest statistics from the United Nations Office for Disaster Risk Reduction (UNDRR) 2021 Special Report estimates the annual losses due to agricultural drought to be USD 6.4 billion in the United States and up to USD 9 billion in Europe [6]. In China, the average annual direct economic losses due to drought from 2006–2017 amounted to RMB 88.230 billion, and the impacted crop area reached up to 169 million hm$^2$ per year [7].

The impacts of agricultural drought are mainly reflected in the reduction of agricultural production, as well as the degradation of forest and grassland [8–10]. The meteorological

drought, caused by factors such as insufficient precipitation, high temperature and high evapotranspiration, can lead to the water supply being less than the requirement, which is manifested as a soil moisture deficit. Then, a soil moisture deficit will affect the normal growth and development of vegetation. For example, drought can lead to the direct death of crops during the germination period. It can directly affect the pollination and fertilization of plants, as well as the growth of fruits during the flowering period of crops, resulting in a crop yield reduction or crop failure. At the same time, the vegetation will change its physiological structure because it cannot obtain water from the soil for growth [11]. Drought indices are an effective means of monitoring agricultural drought, and there are more than 150 drought indices available [3,5]. Among them, drought indices constructed based on meteorological variables, soil moisture and vegetation status can all reflect the possible effects of drought on agriculture from different perspectives.

The drivers of agricultural drought vary from region to region, which indicates that the suitable agricultural drought monitoring indices are diverse for different regions. For example, agricultural production in some regions is mainly sustained by atmospheric precipitation, while others rely on glacial meltwater from upstream, and the influence of human activities (groundwater extraction as well as irrigation, etc.). Suitable drought monitoring indices need to be explored for the specific conditions of different regions to improve the drought monitoring capacity of specific regions [12,13]. Crop yield, as an important indicator of agricultural drought, plays an important role in evaluating the drought index [14,15]. However, long time series of crop yields in the region are difficult to obtain, and the lack of standardization in the collection of statistical data leads to uncertainty in crop yield data.

The net primary productivity (NPP), obtained by using remote sensing satellites over large areas, is strongly correlated with crop biomass [16]. Crop biomass reflects final crop yield [17]. Therefore, NPP could serve as an alternative to crop yield to assess the suitability of drought indices when crop yields are not available [18]. NDVI and NPP indicators can represent the health of vegetation, but the physical significance of NDVI and NPP are considerably different. NDVI represents the greenness and canopy structure. NPP reflects the efficiency of photosynthesis [19]. The relationship between photosynthesis and greenness is unstable, especially under drought stress [20]. The NPP, through the direct effects (e.g., water limitation and heat stress) and indirect effects (e.g., fire and pests) of drought, could lead to yield reduction [21], which means that NPP is sensitive to drought in different seasons and vegetation types [22,23]. The NPP has been extensively used to assess drought indices. Mu et al. developed a new global drought monitoring index, the Drought Severity Index (DSI), using MODIS ET/PET and NDVI data, and used MODIS NPP as a relative indicator of vegetation productivity change to validate the DSI [24]. Wei et al. used the Euclidean distance method and the three-dimensional (3D) P-NDVI-LST to establish the new index Temperature Vegetation Precipitation Dryness Index (TVPDI), which was tested and validated with MODIS NPP [25].

The Shiyang River Basin, located in Gansu Province, China, has the most severe water scarcity and ecological degradation among the three major inland river basins in the Hexi Corridor [26]. The upper reaches, which rely on the surface runoff recharge from glacial meltwater in the northern Qilian Mountains and precipitation in the mountainous areas, are important water conservation areas and have been severely damaged by climate change and irrational development. Midstream and densely populated areas are the main distribution zones of basic farmland and economic industries, which consume most of the upstream runoff water and restricts the amount of water recharge downstream. The lower reaches are surrounded by desert on three sides, with only about 10% being oasis, and the ecological degradation situation is serious. In the past 50 years, the Shiyang River Basin has been massively cleared for farming, and the area of irrigated farmland has been expanding, resulting in an uneven distribution of water resources, a shortage of surface water resources, and an over-exploitation of groundwater, which has caused a series of environmental and ecological problems [27,28]. Therefore, in order to maintain

the economic and social development and ecological stability of the basin and promote the rational exploitation of water resources, it has become particularly important to accurately monitor agricultural droughts in the Shiyang River Basin.

Given that the main drivers of drought in the upper, middle and lower reaches of this basin are different but interlinked, a comparative study of the monitoring capacity of different drought indices in the upper, middle and lower reaches of the Shiyang River Basin will not only provide a direct scientific basis for drought monitoring in this region, but also provide a reference for the selection of drought monitoring indicators for other inland rivers. The main objectives of this study are as follows: (1) to explore the suitability of indices for agricultural drought monitoring in the upper, middle and lower reaches of this basin, taking the Shiyang River Basin as an example, and (2) to analyze the spatial and temporal trends in agricultural drought in the basin from 1982 to 2020. This study uses NPP to represent the effects of agricultural drought on vegetation, assuming that vegetation is only stressed by water conditions, and does not take into account other factors such as disease, pests, etc., as we know that the occurrence of pests and diseases is rare in this region due to its harsh environmental condition [29].

## 2. Materials and Methods

### 2.1. Study Area

The Shiyang River Basin is located in the eastern part of the Hexi Corridor and the northern foot of the Qilian Mountains in Gansu Province, with a geographical location of 101°41′~104°16′ E and 36°29′~39°27′ N (Figure 1). The Shiyang River Basin is a closed inland river basin, far from the sea. It is well known to be short of water due to low precipitation and high evapotranspiration. In the upper reaches of the basin, the average annual temperature is lower than 6 °C, the annual precipitation is 400–600 mm, and the annual evaporation is 700–1200 mm. In the middle reaches, the average annual temperature is lower than 6–8 °C, the annual precipitation is 150–300 mm, and the annual evaporation is 1300–2000 mm. In the lower reaches, the average annual temperature is higher than 8 °C, the annual precipitation is small 150 mm, and the annual evaporation is 2000–2600 mm [30]. Drought in the upper reaches is not only influenced by changes in precipitation, but by changes in glacial snowmelt water. In the middle and lower reaches, irrigation, water transfer and other human activities influence the occurrence of drought disaster [27,31].

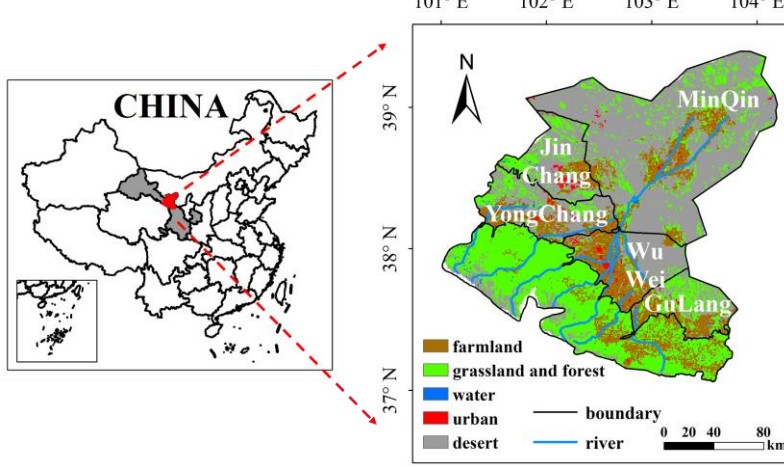

**Figure 1.** Geographical location of the Shiyang River Basin (HRB) in Gansu Province, China, and the distribution of land use and riverways within the Basin.

### 2.2. Datasets and Pre-Processing

2.2.1. Climate Data

TerraClimate provides a global dataset of climate and land surface variables from 1958 to the present (Table 1), with a temporal resolution of 1 month and a spatial resolution



of about 4 km (1/24th of a degree). The accuracy of TerraClimate data is higher than CRU Ts 4.0, as verified by comparison with a large number of station data [32]. This data has been widely used in recent years for ecological and hydrological studies at a global scale [33,34]. In this study, precipitation (P), potential evapotranspiration (PET), actual evapotranspiration (AET), and soil moisture (SM) data are extracted from TerraClimate data for the period 1982–2020. The potential evapotranspiration is calculated using the Penman–Monteith method, which is considered the most suitable method for estimating drought due to the consideration of rich physical variables [35]. Actual evapotranspiration and soil moisture are derived from the output of the water-balance mode (WBM), which is run at monthly time steps and takes into account the interaction between precipitation, potential evapotranspiration, and soil and snow storage [36].

**Table 1.** The information of the data.

| Data | Temporal Resolution | Spatial Resolution | Time Domain | Sources |
|---|---|---|---|---|
| TerraClimate P, PET, AET and SM | 1 month | 4 km | 1982–2020 | http://www.climatologylab.org/terraclimate.html |
| STAR NDVI and BT4 | 1 week | 4 km | 1982–2020 | https://www.star.nesdis.noaa.gov/star/index.php |
| MODIS NPP | 1 year | 500 m | 2000–2020 | https://lpdaac.usgs.gov/products/mod17a3hgfv006/ |
| Land use data | No | 1 km | 2020 | https://www.resdc.cn/ |
| AWC | No | 0.083° | 2000 | https://webmap.ornl.gov/ogcdown/ |

### 2.2.2. Remote Sensing Data

The global VHI product from NOAA Satellite Applications and Research Center (STAR) has similar monitoring results to those of the drought operational monitoring systems GADMFS and USDM (Table 1), which are widely used for agricultural drought monitoring [37]. The weekly Normalized Difference Vegetation Index (NDVI) and Brightness Temperature (BT4), shared by STAR, provide a longer time series (1981-present) data source for vegetation index building [38,39]. Among them, NDVI is calculated by channel 1 and channel 2 of AVHRR, and BT4 is calculated by channel 4 of AVHRR. The NDVI and BT4 datasets with a weekly temporal resolution and a 4 km spatial resolution are generated using smooth filtering (removal of high frequency noise). Since the BT4 is a good indicator of land surface temperature (LST) [40], and has been used as LST for drought monitoring in several studies [41,42], in this study, we use BT4 instead of LST to calculate the drought index. In order to be consistent with the time scale of meteorological data, this study used data obtained from the years 1982–2020, and the maximum value synthesis method to process the weekly data into monthly values; this, at the same time, can well avoid the problem of abruptly small monitoring values due to cloud contamination [43].

The MOD17A3HGF version 6 product is used as the data source for NPP from 2000 to 2020 in this study (Table 1), providing information on annual-scale NPP at a 500 m spatial resolution. Annual NPP is derived from the sum of all 8 days of net photosynthesis (PSN) for a given year. This data is widely used in ecological monitoring [44,45]. MODIS LAI/FPAR, the input data for MODIS NPP, includes a small amount of infrared and near-infrared information. This indicates MODIS NPP and STAR NDVI also take the information of both bands. Shared-band information may cause spurious correlations between the Drought index based on NDVI and NPP. However, MODIS LAI/FPAR represents only a fraction of the MODIS NPP input data and there is a large amount of reanalysis meteorological data [46]. Furthermore, MODIS NPP and STAR NDVI are from different sensors, MODIS, AVHRR, respectively, and the effect of the shared bands is again attenuated. In this study, because of the large number of missing values for MODIS NPP in the lower Shiyang River Basin, this study only analyzes the relationship between the drought index and NPP in the areas where MODIS NPP data is available.

### 2.2.3. Land Use and Other Auxiliary Data

Land use data were obtained from the Resource and Environmental Science and Data Center (Table 1). This data provides annual-scale land use information with a 1 km spatial distribution in China and is widely used in different studies [47]. We selected the land use data of Gansu Province at the time of 2020 and extracted the Shiyang River Basin using Arcgis 10.8. The main land use types in the Shiyang River Basin are farmland, water bodies, desert (classified as unused land in the original data), grassland and forest, and built-up areas (Figure 1). Among them, farmland areas, grassland and forest areas, and desert areas are the subjects of this study. The effective soil water content (AWC) data used to calculate sc-PDSI were obtained from the ORNL DAAC data center on the NASA website (Table 1). It has a spatial resolution of 0.083 degrees and is dated to 2000 years. This data provides accurate and reliable soil information for the calculation of PDSI, as well as a series of modified indices for PDSI [48,49]. All datasets are re-gridded to a 4 km $\times$ 4 km grid resolution, using bilinear interpolation to make data consistent.

### 2.3. Methods

As the complex landscape of the watershed, this study separates the Shiyang River Basin into grassland and forest area, farmland area, and desert area, according to the land use type. This also corresponds to the upper, middle and lower reaches of the watershed by following the topography. Twelve different widely used drought indices, based on meteorological variables (SPEI, sc-PDSI, ESI, MEDDI), soil moisture (SSI, SMA), vegetation status (VCI, TCI, NVWSI, MTVDI), and the multivariate-based integrated drought index (DSI, ISDI), were adopted for the comparison. The detailed process is shown in Figure 2. The full names and abbreviations of the indices are shown in Table 2.

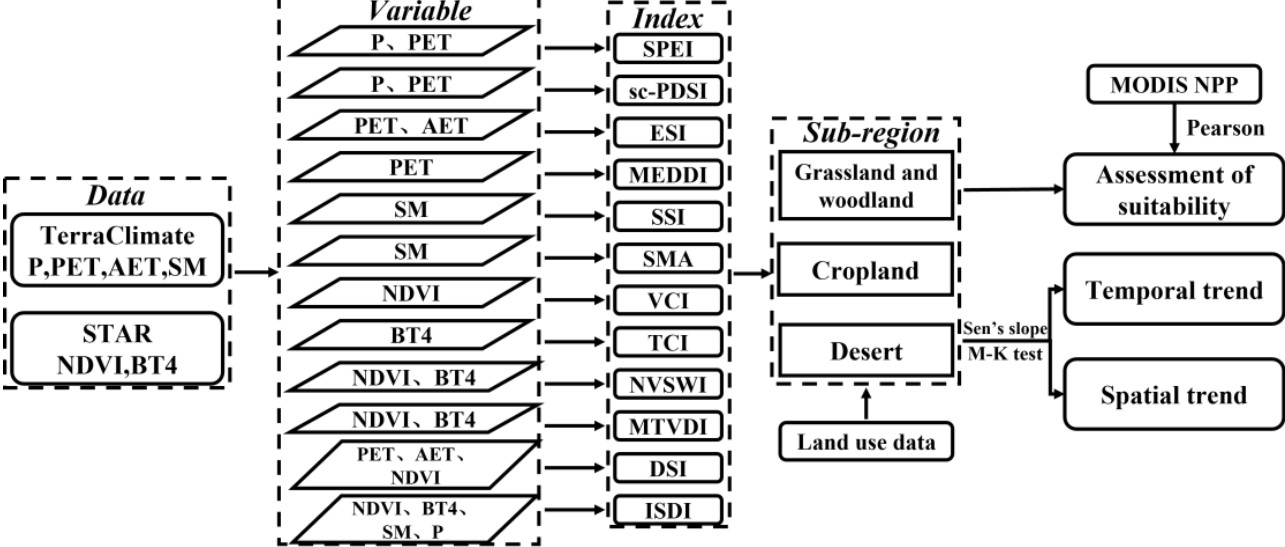

**Figure 2.** Flow chart of calculation and analysis of this study.

**Table 2.** The widely used drought indices.

| Indices | Formula | Variable Explanation | References |
|---|---|---|---|
| SPEI (Standardized Precipitation Evapotranspiration Index) | $f(x) = \frac{\beta}{\alpha}\left(\frac{x-\gamma}{\alpha}\right)^{\beta-1}\left[1+\left(\frac{x-\gamma}{\alpha}\right)^{\beta}\right]^{-2}$ | f(x): Log-Logistic distribution function; $\alpha$, $\beta$ and $\gamma$: Scale parameter, shape parameters and position parameters | [50] |
| sc-PDSI (self-calibrating Palmer Drought Severity Index) | $X_i = pX_{i-1} + qZ_i$ | p and q: Persistence factor; $X_i$ and $X_{i-1}$: Current month PDSI value and previous month PDSI value; $Z_i$: Current month moisture abnormalities | [51] |
| ESI (Evaporative Stress Index) | $ESI = \frac{rET - \overline{rET}}{\delta(rET)}$, $rET = \frac{ET}{PET}$ | ET: Actual evapotranspiration; PET: Potential evapotranspiration | [52] |
| MEDDI (Modified Evaporative Demand Drought Index) | $P = \frac{i-0.33}{n+0.33}$ | P: Cumulative probability; i: The rank of the cumulative quantity after sorting; n: Total number of samples | [53] |
| SSI (Standardized Soil Moisture Index) | $SSI = \frac{SM - \overline{SM}}{\delta(SM)}$ | SM: Soil moisture in the growing season | [54] |
| SMA (Soil Moisture Anomaly) | $SMA = SM - \overline{SM}$ | SM: Soil moisture | [55] |
| VCI (Vegetation Condition Index) | $VCI_i = \frac{NDVI - NDVI_{min}}{NDVI_{max} - NDVI_{min}}$ | NDVI: Normalized vegetation index | [56] |
| TCI (Temperature Condition Index) | $TCI_i = \frac{LST_{max} - LST}{LST_{max} - LST_{min}}$ | LST: Land surface temperature | [41] |
| NVSWI (Normalized Vegetation Supply Water Index) | $VSWI_i = \frac{NDVI}{LST}$ $NVSWI_i = \frac{VSWI - VSWI_{min}}{VSWI_{max} - VSWI_{min}}$ | NDVI: Normalized vegetation index; LST: Land surface temperature | [57] |
| MTVDI (Modified Temperature Vegetation Dryness Index) | $MTVDI = \frac{LST_{max} - LST}{LST_{max} - LST_{min}}$ $LST_{max} = a_1 + b_1 \times NDVI$ $LST_{min} = a_2 + b_2 \times NDVI$ | $a_1$ and $b_1$: Dry-side fitting parameters; $a_2$ and $b_2$: Wet-side fitting parameters; LST Land surface temperature | [58] |
| DSI (Drought Severity Index) | $DSI = \frac{Z - \overline{Z}}{\sigma(Z)}$, $Z = Z_{rET} + Z_{NDVI}$, $rET = \frac{ET}{PET}$ | ET: Actual evapotranspiration; PET: Potential evapotranspiration; Z: Z-score Standardization | [24] |
| ISDI (Integrated Scaled Drought Index) | $ISDI = \frac{1}{6}*Scaled\ NDVI + \frac{1}{6}*Scaled\ LST + \frac{1}{3}*Scaled\ PCP + \frac{1}{3}*Scaled\ SM$ | NDVI: Normalized vegetation index; LST: Land surface temperature; PCP: Precipitation; SM: Soil moisture | [59] |

### 2.3.1. Introduction of Drought Indices

The indices describing agricultural drought can be derived from meteorological variables, soil moisture, and vegetation status, as well as multivariate drought indices. In this study, 12 representative indices were selected for comparison (Table 2). In addition, the growing season is the period when vegetation is most severely affected by drought, so only the growing season period (April–October) is considered in the drought indices calculation.

SPEI, sc-PDSI, ESI and MEDDI are four widely used drought indices based on meteorological variables (Table 2). SPEI is based on the difference between monthly precipitation and potential evapotranspiration, and obtained using normal standardization based on a Log-Logistic probability distribution [50]. SPEI describes long- and short-term droughts using the deviation of water volume in cumulative months, with multiple time scales [60]. Sc-PDSI is based on PDSI with parameters calculated by adjusting to regional characteristics, allowing for more accurate monitoring in the study area [51]. Compared to SPEI, which uses only precipitation and evapotranspiration data, sc-PDSI also incorporates data on soil characteristics, which helps to monitor soil moisture and can effectively describe agricultural drought [15]. ESI is an anomaly in the ratio of actual evapotranspiration to potential evapotranspiration, and this study uses total actual evapotranspiration and potential evapotranspiration during the growing season to calculate the ESI series [52]. EDDI

is ranked using the magnitude of the cumulative values of potential evapotranspiration and is calculated using normal normalization after constructing an empirical cumulative distribution, again with multiple time scales [53]. To facilitate comparison between indices, MEDDI is used which is the negative of EDDI.

The drought indices based on soil moisture calculated with standardized soil moisture data. There are many methods of standardization. Hao et al. used the SPI as an example to fit soil moisture using the Gamma distribution [61], but in the Shiyang River Basin, there are long periods of zero in the TerraClimate soil moisture data, making the Gamma distribution inadequate to fit. Z-score standardization was used to calculate SSI series using total soil moisture during the growing season [54]. SMA is the calculation of the degree of soil moisture deviation from the contemporaneous mean state on a monthly scale [55].

VCI, TCI, NVSWI and MTVDI are commonly used drought monitoring indices based on vegetation status, which is represented by NDVI and LST. VCI and TCI are calculated by normalizing NDVI and LST for the same period separately on a monthly scale, eliminating effects such as season and land cover [41,56]. Under drought stress, NDVI and LST always change in opposite directions. NVSWI is normalized using the ratio of NDVI and LST, and the normalization is intended for comparison with other indices [57]. the trapezoidal relationship between NDVI and LST can establish the wet and dry edges. TVDI uses the relationship between LST and wet and dry edges to determine drought conditions. To ensure that the boundaries of the feature space are representative, the use of TVDI requires that the study area covers all land use types from bare soil to high-density vegetation [58]. MTVDI is a modification of TVDI to make it more suitable for understanding and comparison [62].

DSI and ISDI are drought indices based on the combination of multiple variables from two different models. DSI uses operational satellite remote sensing data as the primary input and is a global index that enhances real-time drought monitoring [24]. DSI uses Z-score to standardize the ratio of actual evapotranspiration to potential evapotranspiration and mean NDVI for the growing season, respectively, to obtain $Z_{rET}$ and $Z_{NDVI}$. The results were summed and normalized again by Z-score. MODIS ET and NDVI data areused to create the DSI. Due to the large number of missing values of MODIS ET in the Shiyang River Basin, a combination of reanalysis ET and remote sensing NDVI is used to calculate DSI. Based on AVHRR NDVI and North American Regional Reanalysis (NARR), ISDI is proposed and applied to agricultural drought monitoring in the U.S. [59]. ISDI assigns different weights to Scaled NDVI, Scaled LST, Scaled PCP, and Scaled SM and combines them linearly based on the correlation analysis with multiple meteorological indices. The weights are determined to be 1/6, 1/6, 1/3 and 1/3, respectively.

### 2.3.2. Correlation Analysis

Pearson's correlation coefficient is used to test the correlation between two variables [63]. In this study, Pearson's correlation coefficient was calculated for the normalized twelve drought indices and the normalized NPP from 2000 to 2020, to determine the similarity between these indices and the NPP, respectively. $|R| < 0.2$ indicates an extremely weak correlation, $0.2 \leq |R| < 0.4$ indicates a weak correlation, $0.4 \leq |R| < 0.6$ indicates a moderate correlation, $0.6 \leq |R| < 0.8$ indicates a strong correlation, and $0.8 \leq |R| \leq 1$ indicates an extremely strong correlation. Statistical significance is provided at the two-sided 5% and 10% level in all cases (*p*-value).

### 2.3.3. Trend Analysis

We use Sen's slope to analyze the trend in the drought index, and the Mann–Kendall method to test the significance of the trend [64,65]. Sen's slope is neither affected by outliers, nor does it need to obey a certain distribution. It has good ability to avoid measurement errors or discrete data. The Mann–Kendal nonparametric test, which is often used in conjunction with Sen's slope; it also does not require the sample to follow a specific distribution and is not disturbed by a few outliers, and has a high degree of quantification.

## 3. Results

### 3.1. Correlation Analysis of Drought Indices and NPP

The twelve drought indices are derived with different assumption and also calculation methods. In addition, the units and magnitude of the indices are different, making it difficult to directly compare and analyze them. In this study, the units of the drought indices and the validation variable NPP are unified by normalization, i.e., the difference between the maximum and minimum values of the series after subtracting the minimum values from the long time series indices. This will make indices more comparable among these indices.

The drought index, based on meteorological variables, has a non-significantly weak correlation with NPP ($0.2 \leq |R| < 0.4$, $p > 0.1$). Most of the meteorological variable-based drought indices are extremely weakly non-significantly correlated with NPP, ($0 \leq |R| < 0.2$, $p > 0.1$) except sc-PDSI, which was weakly non-significantly correlated with NPP ($0.2 \leq |R| < 0.4$, $p > 0.1$) (Figure 3 and Table 3). The four drought indices, based on meteorological variables, had the same trend in interannual variation and the same trend in interannual variation with NPP in most of the periods (Figure 3a,b). The correlations of these four indices with NPP were specifically $R_{SPEI} = 0.146$, $R_{sc-PDSI} = 0.238$, $R_{ESI} = 0.198$, and $R_{MEDDI} = 0.042$ in the grassland and forest area, respectively, and $R_{SPEI} = 0.171$, $R_{sc-PDSI} = 0.242$, $R_{ESI} = 0.153$, and $R_{MEDDI} = 0.165$ for farmland area (Figure 3c,d and Table 3).

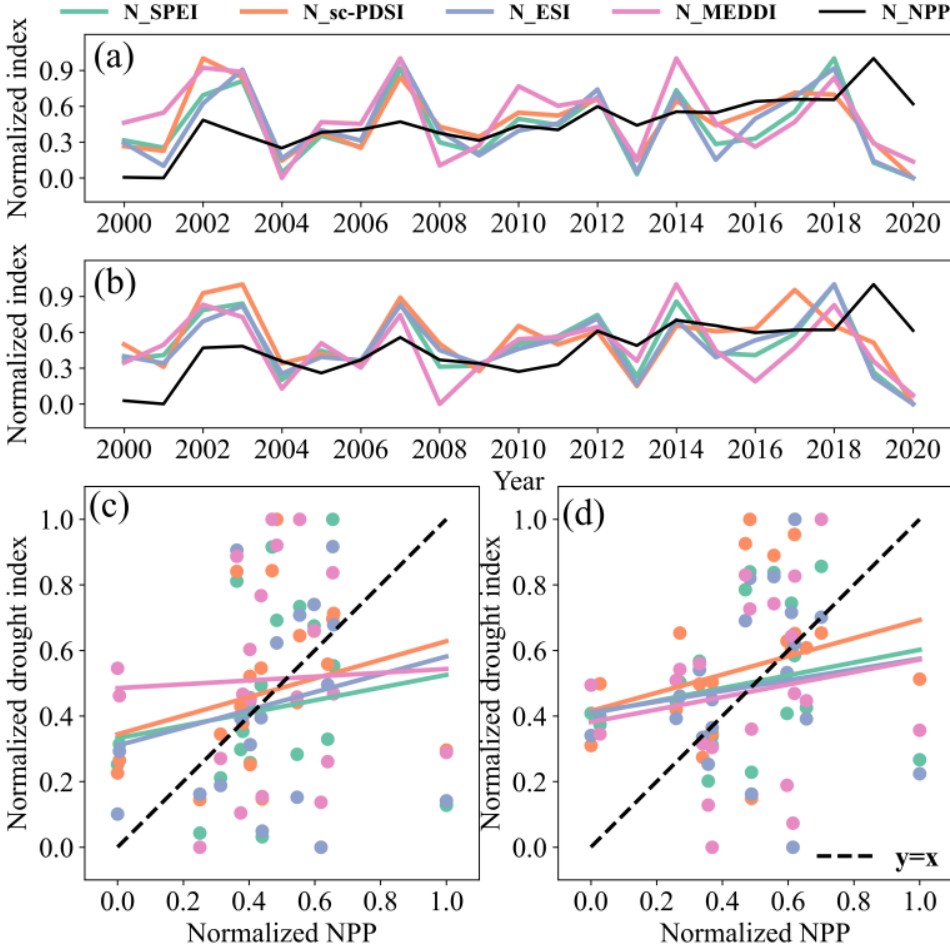

**Figure 3.** Temporal variation in meteorological drought index and NPP during the growing season from 2000 to 2020: (**a**) grassland and forest area and, (**b**) farmland area. The scatter plot distribution of meteorological drought index and NPP: (**c**) grassland and forest area, (**d**) farmland area.

**Table 3.** Correlation coefficients of drought index and NPP in the grassland and forest area, and farmland area, over 2000−2020.

|  | SPEI | sc-PDSI | ESI | MEDDI | SSI | SMA | VCI | TCI | NVSWI | MTVDI | DSI | ISDI |
|---|---|---|---|---|---|---|---|---|---|---|---|---|
| Grassland and forest area | 0.146 | 0.238 | 0.198 | 0.042 | 0.207 | −0.035 | **0.857 **** | −0.056 | **0.623 **** | −0.167 | **0.559 **** | 0.331 |
| Farmland area | 0.171 | 0.242 | 0.153 | 0.165 | **0.482 **** | **0.391 *** | **0.833 **** | −0.158 | **0.657 **** | −0.209 | **0.406 *** | **0.508 **** |

\*\* denotes $p < 0.05$ and \* denotes $p < 0.1$. Higher values of the index indicate greater suitability to monitor agricultural drought. The high R values for each index are shown in bold.

The correlation between the soil moisture-based drought index and NPP is moderately significantly correlated ($0.4 \leq |R| < 0.6$, $p < 0.1$) in the farmland area, which is better than the meteorological variable-based drought index. The correlations of SSI and SMA with NPP are better in the farmland area than in the grassland and forest area, where SSI is moderately significantly correlated with NPP ($0.4 \leq |R| < 0.6$, $p < 0.05$) and SMA is weakly significantly correlated with NPP ($0.2 \leq |R| < 0.4$, $p < 0.1$). In contrast, SSI is weakly non-significantly correlated with NPP ($0.2 \leq |R| < 0.4$, $p > 0.1$). SMA is very weakly non-significantly correlated with NPP ($0 \leq |R| < 0.2$, $p > 0.1$) in the grassland and forest area (Figure 4 and Table 3). The interannual trends of these two indices are the same in the farmland area (Figure 4b). In the grass and forest area, the same interannual trends are monitored for both indices, except in 2015–2016 (Figure 4a). Before 2013, the interannual trends in NPP and the soil moisture-based drought index are the same. After 2013, the soil moisture-based drought index show an increasing trend. The changes in NPP always lag behind the soil moisture-based drought index, with a lag of one growing season (Figure 4a,b). The correlation coefficients between the two indices and NPP showed $R_{SSI} = 0.207$ and $R_{SMA} = -0.035$ in the grassland and forest areas, and $R_{SSI} = 0.482^{**}$ and $R_{SMA} = 0.391^*$ in the farmland areas, respectively (Figure 4c,d and Table 3).

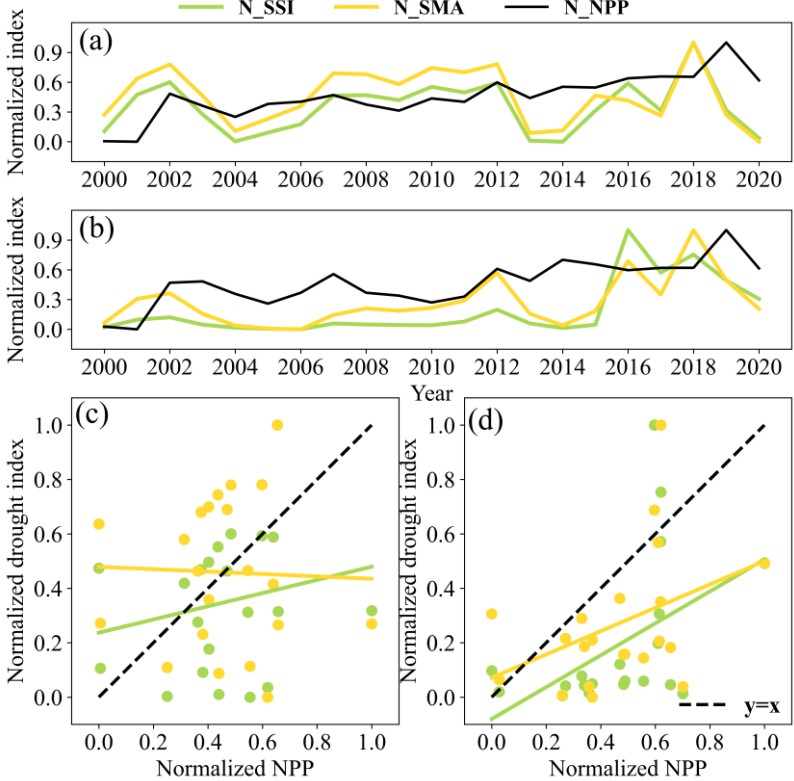

**Figure 4.** Temporal variation in soil moisture-based drought index and NPP during the growing season from 2000 to 2020: (**a**) grassland and forest area, (**b**) farmland area. The scatter plot of soil moisture drought index and NPP from 2000 to 2020: (**c**) grassland and forest area and, (**d**) farmland area.

The drought index based on vegetation status is extremely strongly significantly correlated with NPP ($0.8 \leq |R| \leq 1$, $p < 0.05$). VCI has the highest correlation with NPP, reaching an extremely strong significant correlation ($0.8 \leq |R| \leq 1$, $p < 0.05$). This is followed by NVSWI, which reached a strong significant correlation ($0.6 \leq |R| < 0.8$, $p < 0.05$). TCI is extremely weakly non-significantly correlated with NPP ($0 \leq |R| < 0.2$, $p > 0.1$). MTVDI is weakly non-significantly correlated with NPP in the farmland area ($0.2 \leq |R| < 0.4$, $p > 0.1$) and extremely weakly non-significantly correlated in the grassland and forest area ($0 \leq |R| < 0.2$, $p > 0.1$) (Figure 5 and Table 3). The interannual variation in the four indices is quite different (Figure 5a,b). For most of the periods, VCI and NVSWI show the same trend in interannual variation, while TCI and MTVDI show the same trend in interannual variation only in the grassland and forest area. In addition, the interannual variation trend in NPP is the same as VCI and NVSWI, and opposite to TCI and MTVDI in most periods. The correlation between VCI and NVSWI and NPP was shown as $R_{VCI} = 0.857**$ and $R_{NVSWI} = 0.623**$ in the grassland and forest area, and $R_{VCI} = 0.833**$ and $R_{NVSWI} = 0.657**$ in the farmland area, respectively. The correlation between TCI and MTVDI and NPP show that $R_{TCI} = -0.056$ and $R_{MTVDI} = -0.167$ in the grassland and forest area, and $R_{TCI} = -0.158$ and $R_{MTVDI} = -0.209$ in the farmland area, respectively (Figure 5c,d Table 3).

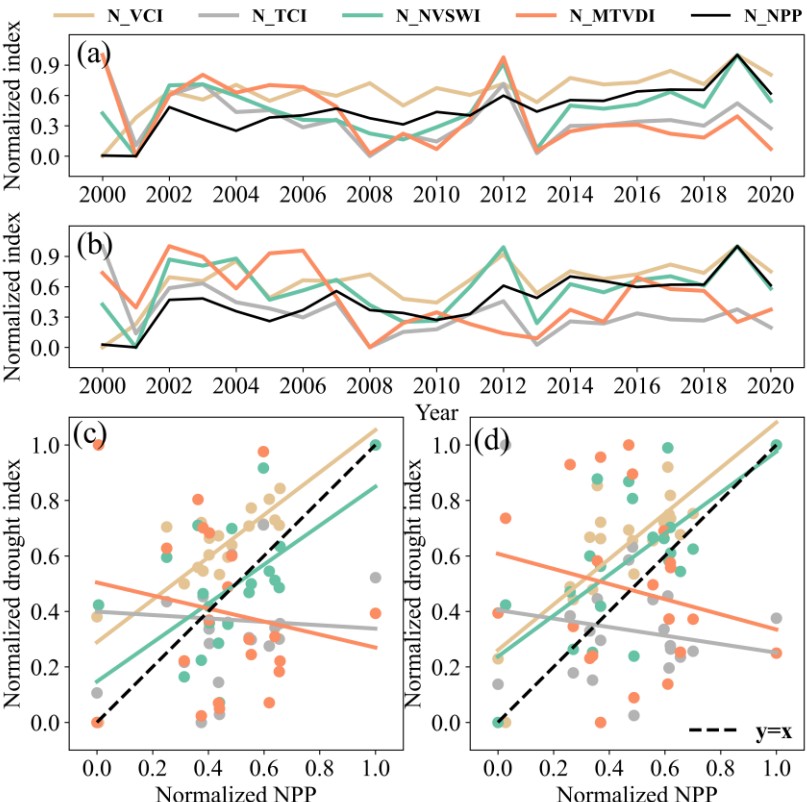

**Figure 5.** Temporal changes of vegetation status-based drought index and NPP during the growing season from 2000 to 2020: (**a**) grassland and forest area, (**b**) farmland area. The scatter plot distribution of vegetation status-based drought index and NPP over 2000–2020: (**c**) grassland and forest area and (**d**) farmland area.

The integrated drought index is moderately significantly correlated with NPP ($0.4 \leq |R| < 0.6$, $p < 0.1$), which is better than the drought index based on meteorological variables and soil moisture. The DSI is moderately significantly correlated with NPP ($0.4 \leq |R| < 0.6$, $p < 0.1$). The ISDI is moderately significantly correlated with NPP in the farmland areas ($0.4 \leq |R| < 0.6$, $p < 0.05$) and weakly non-significantly correlated in the grassland and forest areas ($0.2 \leq |R| < 0.4$, $p > 0.1$) (Figure 6 and Table 3). The interannual trends

in the two combined drought indices are opposite only in 2016–2017 (Figure 6a,b). The interannual trends in DSI and ISDI with NPP are opposite in the grassland and forest area for 2017–2019, and in the farmland area for 2008–2010, 2015–2016 and 2017–2019. The correlation between the two integrated drought indices and NPP is shown as $R_{DSI} = 0.406*$ and $R_{ISDI} = 0.508**$ in the farmland area, and $R_{DSI} = 0.559**$, $R_{ISDI} = 0.331$ in the grassland and forest area, respectively (Figure 6c,d and Table 3).

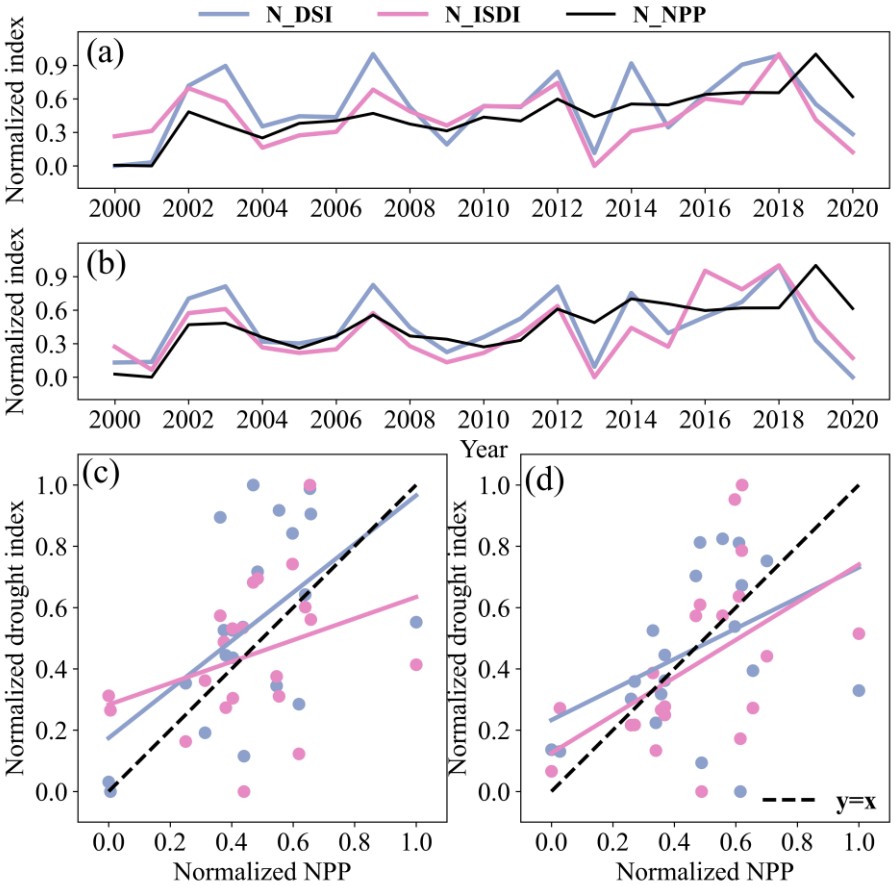

**Figure 6.** Temporal variation in integrated drought index and NPP during the growing season from 2000 to 2020: (**a**) grassland and forest area and (**b**) farmland area. The scatter plot distribution of integrated drought index and NPP: (**c**) grassland and forest area and (**d**) farmland area.

*3.2. Temporal Trends of Agricultural Drought*

Trends in the development of agricultural drought in the study area are analyzed based on the four drought indices (VIC, NVSWI, DSI and ISDI), which are identified as suitable indices for agricultural drought in previous section. The study shows that there is a trend of a decrease in agricultural drought in the study area, which is represented by an increasing in these indices (VIC, NVSWI, DSI and ISDI) (Figure 7, Table 4). In the grassland and forest area, VCI shows a significant ($p < 0.05$) increasing trend, with a rate of change of 0.06/10a. NVSWI shows a non-significant decreasing trend, with a rate of change of $-0.01$/10a. DSI shows a significant ($p < 0.1$) increasing trend, with a rate of change of 0.23/10a. ISDI shows a non-significant increasing trend, with an increasing trend approximately equal to 0. In the farmland area, VCI shows a significant ($p < 0.05$) increasing trend, with a rate of change of 0.09/10a. NVSWI shows a significant ($p < 0.1$) increasing trend, with a rate of change of 0.05/10a. DSI shows a significant ($p < 0.05$) increasing trend, with a rate of change of 0.43/10a. ISDI shows a non-significant increasing trend, with a rate of increase equal to 0. In the desert area, VCI shows a significant ($p < 0.1$) increasing trend, with a rate of change of 0.04/10a and NVSWI shows a non-significant increasing trend, with a rate of change of 0.01/10a. DSI shows a significant ($p < 0.1$) increasing trend,

with a rate of change of 0.23/10a. ISDI shows a significant ($p < 0.05$) decreasing trend, with a rate of decrease of $-0.03/10a$.

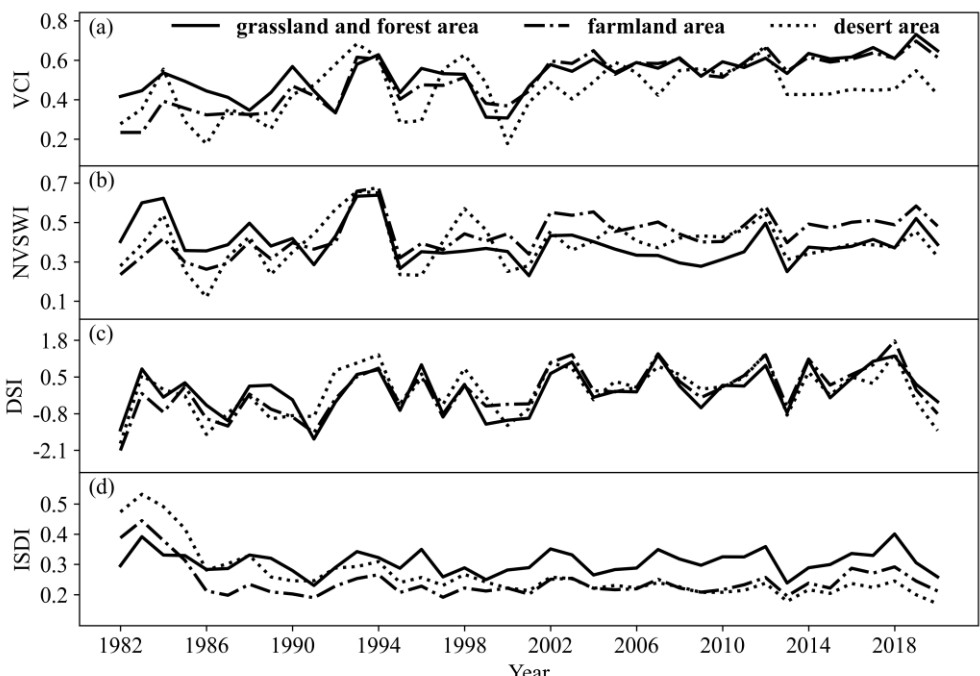

**Figure 7.** Temporal changes in (**a**) VCI, (**b**) NVSWI, (**c**) DSI and (**d**) ISDI in the grassland and forest area, and farmland area and desert area, over 1982–2020.

**Table 4.** Temporal trends in VCI, NVSWI, DSI and ISDI in the grassland and forest area, and farmland area and desert areas, over 1982–2020.

|  | **VCI** | **NVSWI** | **DSI** | **ISDI** |
|---|---|---|---|---|
| Grassland and forest area | 0.06/10a ** | −0.01/10a | 0.23/10a * | 0.00/10a |
| Farmland area | 0.09/10a ** | 0.05/10a * | 0.43/10a ** | 0.00/10a |
| Desert area | 0.04/10a * | 0.01/10a | 0.23/10a * | −0.03/10a ** |

** denotes $p < 0.05$ and * denotes $p < 0.1$. Higher values of the index indicate weaker agricultural drought.

### 3.3. Spatial Distribution of Agricultural Drought Trend

The spatial variation in the four indices (VIC, NVSWI, DSI and ISDI) indicates a significant slowdown in agricultural drought in the Shiyang River Basin (Figure 8). The VCI increases significantly in the agricultural area, as well as in the southern side of the grassland and forest area near the glacier, with trends ranging from approximately 0.015 to 0.03/a. The area of significant increase is less in the desert area, with a trend between approximately 0 and 0.015/a (Figure 8a). NVSWI monitored a significant increasing trend in most of the farmland area, with a trend between approximately 0.015 and 0.03/a. A significant decreasing trend in NVSWI was monitored in a small part of the grassland and forest area, especially in the upper grassland and forest area bordering the farmland area, with a trend of between approximately −0.015 and 0/a (Figure 8b). The distribution of spatial variation in DSI and VCI is similar, with a significant increase in most of the area and an increasing trend of approximately 0.045/a or more (Figure 8c). ISDI significantly increases in a small part of the farmland area, which is mainly in the middle stream, with a trend of between approximately 0 and 0.015/a. In addition, a significant decreasing trend in ISDI was monitored in the northern part of the downstream desert zone, with a trend of between approximately −0.015 and 0/a (Figure 8d).



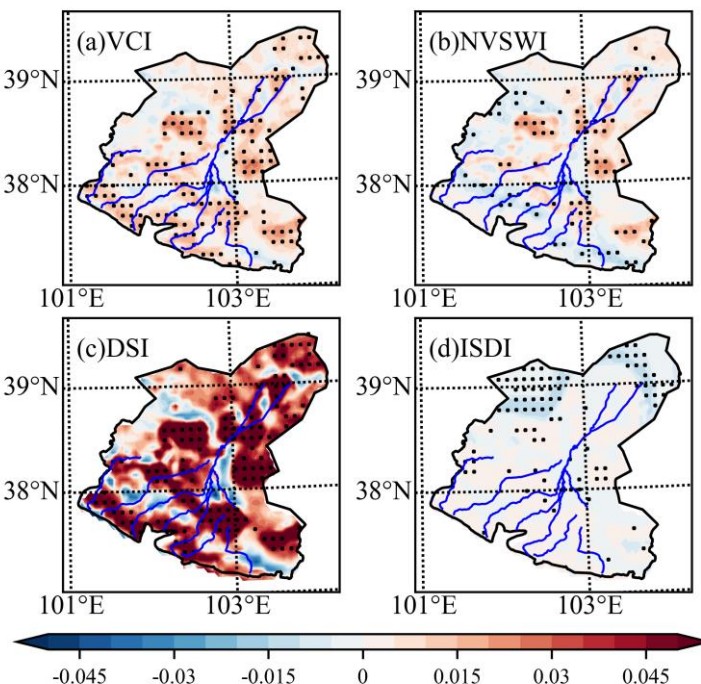

**Figure 8.** Spatial distribution of trends: (**a**) VCI, (**b**) NVSWI, (**c**) DSI, and (**d**) ISDI, over 1982–2020. The black dots represent significance above 95%.

## 4. Discussion

### 4.1. Suitability Indices for Agricultural Drought Monitoring in the Shiyang River Basin

VCI, NVSWI and the two integrated drought indices are suitable for monitoring agricultural drought in the Shiyang River Basin. The drought index based on meteorological variables is affected by meteorological conditions such as water scarcity and high evapotranspiration. The change from meteorological environmental background to vegetation growth and development is a complex process, resulting in some areas where drought indices based on meteorological variables do not directly respond to the status of agricultural drought [66]. With global warming and accelerated glacier shrinkage, the melting of glaciers in the upper reaches of the Shiyang River basin has allowed more runoff to be used at lower elevations to compensate for the effects of meteorological drought on vegetation [67]. In addition, more than 90% of the surface water in the Shiyang River basin is allocated to agricultural land, while large amounts of groundwater are extracted. In the event of a major drought, water catchment transfers from outside areas mitigate the effects of drought on crops [26]. Soil moisture is the indicator most closely related to vegetation growth [68]. The monitoring capability of soil moisture-based drought indices in agricultural areas has been improved compared to that of drought indices based on meteorological variables. However, the sensitivity differs between soil moisture at different depths and different types of vegetation [69,70]. The data quality of soil moisture is another factor that has been limiting the development of drought monitoring [71], especially in the arid and semi-arid regions of northwest China, which cannot rely excessively on soil moisture to monitor agricultural drought. The results showed that VCI and NVSWI, two indices with considering NDVI, performed better in monitoring agricultural drought in the Shiyang River Basin. NDVI is one of the important parameters reflecting crop growth and nutrient information. Adding LST is not suitable for monitoring agricultural drought in the Shiyang River Basin, which is similar to Wei's study [72]. The integrated drought index has a significantly higher monitoring capacity due to the consideration of more variables, most notably NDVI, and has a good performance in different areas of the Shiyang River Basin.

*4.2. The Contrasting Trend between Meteorological and Agricultural Drought*

Agricultural drought in the study area tended to slow down, but drought indices based on meteorological variables showed a trend of increasing drought (Figure A1, Table A1). In the grassland and forest area, drought indices based on meteorological variables has a generally drier trend. In the farmland area, no significant trends were monitored for the drought indices based on meteorologically related variables. In the desert area, the drought indices based on meteorological variables monitored a trend of gradually increasing drought severity. These manifestations, contrary to the actual agricultural drought changes, are mainly due to the improvement of water resource management and utilization efficiency, in order to reduce the impact of meteorological conditions on agricultural drought. In the upstream area, abundant water resources, brought by the melting of the glaciers, keep the soil moist and the soil moisture maintains the normal water demand of vegetation [73]. In the midstream region, the role of human activities dominates the changes in soil moisture and vegetation, and irrigation and water allocation are the main means for local crops to obtain water resources. With optimal and orderly field management by the government, crops are less affected by meteorological conditions [27]. Downstream has received attention from authorities and scholars for many years, and thus the vegetation cover and growth status have been improved thanks to a series of ecological projects, such as water harvesting, water transfer and groundwater extraction, implemented in the downstream [28]. By comparing trends in drought indices based on meteorological variables and agricultural droughts, especially over different time windows, it may be useful to analyze the causes of agricultural drought formation and even to quantify the contribution of human activities in mitigating agricultural droughts.

*4.3. The Importance of Runoff Information in Agricultural Drought Monitoring for Inland River Basin*

Drought indices that consider runoff would be more effective in monitoring agricultural drought in the Shiyang River Basin, a water management intensive region. In the closed inland river, precipitation is not the only water resource, but runoff from melting glaciers in the upper reaches and groundwater can also meet most of the agricultural needs of the middle and lower reaches [74]. During the flood season, the flow of Shiyang River is mainly influenced by the precipitation in the Qilian Mountains, and the drought indices based on meteorological variables can reflect agricultural drought to some extent. During the non-flood period, runoff is supplemented by the melting of snow and ice formed by the upstream glaciers; at this time, the drought index based on meteorological variables cannot reflect the actual water deficit [75]. Precipitation is low in the middle and lower reaches in the Shiyang River Basin and potential evapotranspiration is high; thus, vegetation is mainly dependent on moisture replenished by runoff, which greatly reduces the dependence of agricultural drought on precipitation. Agricultural drought, as a type of drought severely affected by human activities, is more closely related to water resource management, such as irrigation and reservoirs; hydrological process models that take these factors into account have been well validated for monitoring agricultural droughts [76]. The vast majority of these water sources come from runoff, which reflects the efficiency of water management and use in inland river basins and the integration of natural and human factors. [77,78]. Among the runoff-related drought indices, PDSI has been extensively validated in global and regional-scale drought monitoring [79,80]. The use of SRI, SWSI, and RDAI may further enhance the capability of agricultural drought monitoring in the Shiyang River Basin [81]. In fact, the practice of incorporating the runoff drought index to monitor agricultural drought has been applied in the operational agricultural drought monitoring system USDM in the United States [82].

*4.4. The Imperative of Constructing Integrated Drought Indices*

The key to constructing a comprehensive drought index is the selection of variables and the construction of the model. The selection of variables representing the drivers

of regional agricultural drought can significantly improve the monitoring ability of the index. In the results of this study, TCI and MTVDI derived with LST were weakly or even very weakly correlated with NPP after normalization in the Shiyang River Basin sub-region. The spatial and temporal variations in NVSWI and ISDI were significantly different from VCI and DSI in the desert and grassland, and forest areas. The LST was not suitable to be added as an input in monitoring the drought index in the Shiyang River Basin. The monitoring ability of the index may be controlled by only some key variables, and adding more parameters may introduce more errors [83,84]. For multivariate based composite drought indices, the construction methods are complex and the most commonly used methods include linear combination, joint distribution, and principal component analysis [1]. Considering regional heterogeneity, such as climatic conditions, vegetation types, and human activities, can enhance the monitoring effectiveness of the drought index [76,85]. There is a significant cumulative and lagged relationship among meteorological conditions, soil moisture, and vegetation status. The severe agricultural drought can be triggered by long-term or growth-critical environmental stresses. The time scale and lag effects of variables need to be considered when constructing a comprehensive drought index [86,87]. In addition, the use of machine learning and deep learning can resolve the complex and ambiguous nonlinear relationships among different variables, which is a hot research topic for constructing integrated drought indices [1].

*4.5. Practical Implication and Limitations*

This study revealed that the vegetation status-based and the integrated drought indices have a better performance than meteorological variables or soil moisture-based drought indices in this inland river basin with intensive water management. This finding is essential for other regions with intensive water management, because the meteorological variables-based drought indexes are not suitable for drought management in this region. Meanwhile, the identified suitable drought indices are based on the widely available climate and remote sensing data. All of them can be easily obtained for the agricultural drought monitoring. While NPP is a good indicator of crop yield production, it would be better to obtain the yield production to validate these drought indices.

**5. Conclusions**

In this study, the suitability of drought indices for the agricultural drought monitoring of an inland river (the Shiyang River Basin) was explored, based on the widely used drought indices. Then, the spatial–temporal trend in agricultural drought was analyzed in the study region. We reached with following conclusions:

(1) Drought indices based on meteorological variables are not suitable for monitoring agricultural drought in the Shiyang River Basin. The drought indices based on meteorological variables are extremely weakly non-significantly correlated with NPP ($0 \leq |R| < 0.2$, $p > 0.1$). The drought indices based on meteorological variables can only partially reflect the interannual variation trend in NPP.

(2) The drought index based on soil moisture is more capable of monitoring in farmland area than the drought index based on meteorological variables. The drought index based on soil moisture can be moderately significantly correlated with NPP in the farmland area ($0.4 \leq |R| < 0.6$, $p < 0.1$), and the interannual variation in the soil moisture-based drought index lags behind the NPP, which can predict the change in agricultural drought in advance.

(3) VCI and NVSWI can accurately monitor the variation in NPP, and the indices constructed using LST reduce the monitoring ability. NPP is extremely strongly significantly correlated with VCI ($0.8 \leq |R| \leq 1$, $p < 0.05$) and strongly significantly correlated with NVSWI ($0.6 \leq |R| < 0.8$, $p < 0.05$). TCI and MTVDI are not strongly non-significantly correlated with NPP ($0 \leq |R| < 0.4$, $p > 0.1$).

(4) The integrated drought index improves the ability and stability of monitoring compared with the index considering only single factors. The integrated drought index

can achieve a moderate significant correlation with NPP ($0.4 \leq |R| < 0.6$, $p < 0.1$), which is better than the drought index based on meteorological variables and soil moisture.

(5) The monitoring of the spatial and temporal changes in the suitability indices VCI, NVSWI, DSI and ISDI showed that the drought index based on meteorological variables showed a trend of increasing drought, while agricultural drought slowed down during 1982–2020 in the Shiyang River Basin. This is mainly due to the improvement of water management and utilization efficiency within the basin.

This study provides the basis for index selection in agricultural drought monitoring in the Shiyang River Basin. Meanwhile, the drought index assessment framework in this study can be applied to agricultural drought monitoring in other inland river basins.

**Author Contributions:** Conceptualization, S.M.; methodology, S.M. and W.L.; software, W.L.; validation, S.M., W.L. and Y.G.; formal analysis, S.M. and W.L.; investigation, W.L.; resources, W.L.; data curation, W.L.; writing—original draft preparation, S.M. and W.L.; writing—review and editing, W.L., S.M. and K.F.; visualization, W.L.; supervision, S.M. and W.L.; project administration, W.L., L.L. and M.T.; funding acquisition, S.M. All authors have read and agreed to the published version of the manuscript.

**Funding:** This research was funded by the National Key Research and Development Program of China (No. 2017YFE0119100) and "Hundred Talent Program" of the Chinese Academy of Sciences (Y729G01002).

**Data Availability Statement:** Not applicable.

**Acknowledgments:** The principal author is extremely indebted to all data providers and their staff for establishing and upholding the sites used in the current study. We also wish to express our gratitude to Key Laboratory of Desert and Desertification, Northwest Institute of Eco-Environment and Resources, Chinese Academy of Sciences, China, for providing essential infrastructure without which this work would not have been possible. We would like to thank the anonymous reviewers for their precious attention.

**Conflicts of Interest:** The authors declare no conflict of interest.

## Appendix A

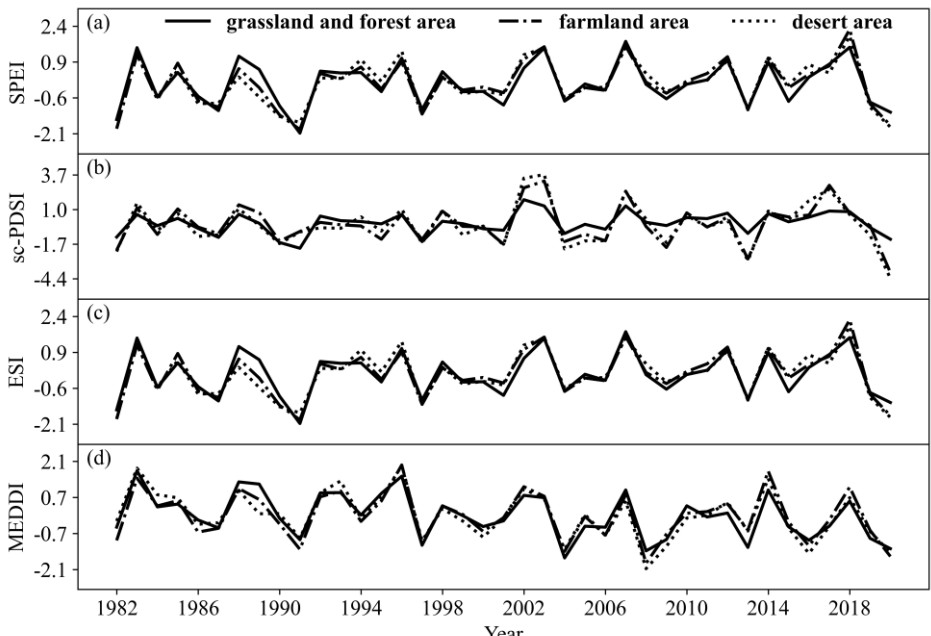

**Figure A1.** Temporal changes (**a**) SPEI, (**b**) sc-PDSI, (**c**) ESI and (**d**) MEDDI in the grassland and forest area, farmland area and desert area over 1982–2020.

**Table A1.** Temporal trends of SPEI, sc-PDSI, ESI and MEDDI in the grassland and forest area, farmland area and desert area, over 1982–2020.

|  | SPEI | sc-PDSI | ESI | MEDDI |
|---|---|---|---|---|
| Grassland and forest area | 0.10/10a | 0.13/10a | 0.10/10a | −0.32/10a ** |
| Farmland area | 0.16/10a | 0.02/10a | 0.16/10a | −0.1710a |
| Desert area | 0.22/10a | 0.16/10a | 0.22/10a | −0.30/10a ** |

** denotes $p < 0.05$. Higher values of the index indicate weaker agricultural drought.

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
