# Peer review of "The Suitability Assessment of Agricultural Drought Monitoring Indices: A Case Study in Inland River Basin"

_agronomy, doi:10.3390/agronomy13020469_

Round 1
Reviewer 1 Report
The authors presented a good paper which compares the accuracy of a number of remote sensing and meteorological methods of mapping and monitoring drought indices in a river basin in China. The scientific merits of the paper are of a high quality. However, the authors introduced results and discussion for two concepts whose background was not given, nor the justification for the study made. This sudden introduction of concepts affect the quality of the paper, including the relevance of the topic. The authors should remove the concepts and remain with the core of the paper which has already been discussed in the background.
Reviewer 2 Report
the authors attempt to evaluate the suitability assessment of agricultural drought monitoring through several known indices. it is an interesting topic in agricultural science. However, there are some issues that should be clarified before publication in Agronomy.
1- the originality of the work is not clear. please highlight the novelty of the manuscript
2- you found that VCI, NVSWI (Vegetation status-based drought indices), DSI, and ISDI are the best indices for assessing agricultural drought. my question is: are these indices generalizable? what about the reproducibility of the methodology adopted?
3- please provide a clear discussion of the practical implication and limitations of the proposed methodology.
Reviewer 3 Report
The purpose and scope of the work were correctly defined. The literature review in the introduction to the article is correct and includes current research related to monitoring and measuring drought in agriculture. The review of measurement indexes is very rich and well introduces the reader to the subject of the article.
The methodology of twelve indicators is very well presented in the table. I have no comment on this part.
The result of the analysis for the study area is presented in the form of tables and graphs - which makes it easier to understand the results.
Conclusions are correct.
Detailed notes:
- in the caption of figure 1, the link to the website suggests moving to the list of literature.
- similarly in lines 135, 166, 182, 190
Reviewer 4 Report
The paper examined how indices work to monitor agricultural drought in the upper, middle, and lower parts of this basin by taking the Shiyang River Basin as a study area. It also examined how the agricultural drought has changed over time and space in the basin from 1982 to 2020. The paper is well-written and includes a lot of important information that will help researchers in the region and worldwide. I think it needs minor revision.
1. The authors need to discuss the importance of the paper on a regional and global scale.
2. The study area sections are very briefly discussed. I recommend the authors discuss other details of the study area, such as climate, rainfall, and topography, using bar graphs and images.
3. Introduction section can be strength by using suggested sites: https://www.sciencedirect.com/science/article/abs/pii/S0022169422003900;https://www.sciencedirect.com/science/article/abs/pii/S0022169420308180; https://www.mdpi.com/2071-1050/14/16/10430
4. Please include the list of abbreviations.
5. The quality and resolution of all figures used, including the graphical abstract, are poor. The text written inside the figure is difficult to read. I recommend that the author increase the resolution of all figures to 300 dpi or higher.
6. I encourage the author to read the entire text carefully and correct all spelling and grammatical errors.
Round 2
Reviewer 2 Report
the paper is ready for publication